# The “Invisible Enemy” SARS-CoV-2: Viral Spread and Drug Treatment

**DOI:** 10.3390/medicina58020261

**Published:** 2022-02-10

**Authors:** Alina Tanase, Aniko Manea, Alexandra Denisa Scurtu, Lavinia Melania Bratu, Doina Chioran, Alina Dolghi, Iren Alexoi, Hazzaa AAbed, Voichita Lazureanu, Cristina Adriana Dehelean

**Affiliations:** 1Department of Management, Legislation and Communication in Dentistry, Faculty of Dental Medicine, “Victor Babes” University of Medicine and Pharmacy, Eftimie Murgu Square No. 2, 300041 Timisoara, Romania; tanase.alina@umft.ro; 2Department of Neonatology and Childcare, Faculty of Medicine, “Victor Babes” University of Medicine and Pharmacy, Eftimie Murgu Square No. 2, 300041 Timisoara, Romania; manea.aniko@umft.ro; 3Department of Toxicology and Drug Industry, Faculty of Pharmacy, “Victor Babes” University of Medicine and Pharmacy, Eftimie Murgu Square, No. 2, 300041 Timisoara, Romania; dolghi.alina@umft.ro (A.D.); contact@drirenalexoi.ro (I.A.); cadehelean@umft.ro (C.A.D.); 4Department of Psychology, Faculty of Medicine, “Victor Babes” University of Medicine and Pharmacy, Eftimie Murgu Square No. 2, 300041 Timisoara, Romania; 5Department of Dento-Alveolar Surgery, Faculty of Dental Medicine, “Victor Babes” University of Medicine and Pharmacy, 9 Revolutiei 1989 Ave., 300070 Timisoara, Romania; chioran.doina@umft.ro; 6Department of Plastic Surgery, Faculty of Medicine, “Victor Babes” University of Medicine and Pharmacy, Eftimie Murgu Square No. 2, 300041 Timisoara, Romania; haza_a_dr@hotmail.com; 7Department of Infectious Diseases II, Faculty of Medicine, “Victor Babes” University of Medicine and Pharmacy, Eftimie Murgu Square No. 2, 300041 Timisoara, Romania; lazureanu.voichita@umft.ro

**Keywords:** SARS-CoV-2, COVID-19, viral spread, treatment, monoclonal antibodies

## Abstract

Nowadays, severe acute respiratory syndrome coronavirus 2 (SARS-CoV-2) infection has become the main subject of the scientific medical world and all World Organizations, causing millions of deaths worldwide. In this review, we have highlighted the context of the Coronavirus disease 2019 (COVID-19) pandemic, how the virus spreads, the symptoms and complications that may occur, and, especially, the drug treatment of viral infection, with emphasis on monoclonal antibodies. While well-known strains such as Alpha, Beta, Gamma, and, especially, Delta have shown an accelerated transmission among the population, the new Omicron variant (discovered on 24 November 2021) indicates more significant infectiousness and the poor efficacy of monoclonal antibody therapy due to mutations on the spike protein receptor-binding domain. With these discoveries, the experiments began, the first being in silico and in vitro, but these are not enough, and in vivo experiments are needed to see exactly the cause of neutralization of the action of these drugs. Following the documentation of the latest medical and scientific research, it has been concluded that there are many chemical molecules that have the potential to treat SARS-CoV-2 infection, but more detailed clinical trials are needed for their use in therapy. In addition, it is important to consider the structure of the viral strain in the administration of treatment.

## 1. Context of the COVID-19 Pandemic

In recent years, viral diseases have become a major threat to people around the world, with more and more outbreaks of viral diseases reported, such as the H1N1 influenza in 2009, the Middle East respiratory syndrome coronavirus (MERS-CoV) disease in 2012, and Ebola and Zika virus infections in 2013 and 2015, respectively [1].

The origin of the pandemic caused by the new coronavirus had as a starting point the Chinese city of Wuhan, the first cases being detected in December 2019. At the end of 2019, a series of atypical respiratory infections devastated the city, finding that the agent incriminated by the infection is a new type of coronavirus, called severe acute respiratory syndrome coronavirus 2 (SARS-CoV-2), belonging to the *Coronaviridae* family. Thus, the disease was named Coronavirus disease 2019 (COVID-19) by World Health Organization (WHO) [2,3]. Since then, the virus has shown evidence of human-to-human transmission, with the rate of infection rising sharply in mid-January, with the disease being reported in all countries around the world [4,5,6]. At the end of February 2020, Italy reported a significant increase in COVID-19 cases, most of which were identified in the northern part of the country. The pandemic in northern Italy also had a real psychological impact on those working in the field of health care. The psychological impact caused by the emergence of COVID-19 on the medical staff operating in Italy was, in some places, even stronger than that reported in China, the place where this pandemic broke out [7].

In March 2020, cases of COVID-19 were reported in every Member State of the European Union, which led the WHO to declare on 11 March 2020 the appearance of COVID-19 as a global pandemic [8]. In less than two days, on 13 March 2020, the World Health Organization declared Europe the epicenter of the coronavirus pandemic. At that time, the daily number of cases detected in Europe was higher than that reported in China at the height of the pandemic [9]. In the first month of 2022, WHO summarized the results of the infection. The contamination figure in January exceeded 360 million (MLN), resulting in more than 5,600,000 deaths (1.55% of the total infected). Figure 1 shows the number of involvements and deaths recorded by January of this year; the situation is organized by WHO regions. So far, the European region had the most confirmed cases, more than 138 MLN, with a mortality rate of 1.55% of the total contamination, followed by the Americas with about 132 MLN and a mortality rate of 1.88%. In South-East Asia, the mortality percentage is lower, with 0.45 %, in comparison with the Americas. The African region had fewer confirmed cases, but the mortality rate was higher (2.03%). In the Western Pacific region, the situation is more optimistic, with a mortality rate close to 1% (1.11%) [10].

In the last year, more and more variants of the SARS-CoV-2 virus have been discovered, including B.1.1.7 (Alpha) in the United Kingdom, B.1.351 (Beta) in South Africa, B.1.1.28 (Gamma), B.1.617. 2 (Delta) in Brazil, and the most recent, which has proven to be the most transmissible, B.1.1.529, the Omicron variant in South Africa [11,12].

Specifically, in November 2021, the WHO informed the population of the appearance of a new variant of SARS-CoV-2, called Omicron. With the discovery of this strain, the medical and scientific world began a journey in the exact knowledge of structure. Concerns began to arise when it was shown that the Omicron variant carries a very large number (82) of mutations, 32 mutations, on spike protein (S), which is one of the most important, being the main antigenic target of antibodies. Up until then, the contagiousness and severity of the Delta variant, which proved to have only 5 mutations in the spike protein, were well known [13].

The emergence of the new strain has raised questions in the medical world; there are suspicions that established drug therapy, especially the monoclonal antibodies tested so far, is not effective in treating the new variant of severe acute respiratory syndrome coronavirus 2.

As is well known, the action of monoclonal antibodies is achieved by binding to the RBD (S protein receptor-binding domain). Therefore, any mutation in the spike protein may lead to the ineffectiveness of the monoclonal antibodies [13].

First of all, this review focuses on information about the transmission of SARS-CoV-2 infection in the population, the symptoms and complications that may occur, the notions needed to understand the disease, and then the classic drug treatment, especially monoclonal antibodies, focusing at this time on the new Omicron variant and on the effectiveness of the existing therapy on this strain. Finally, we also discuss the implications of malpractice in the context of the SARS-CoV-2 crisis.

## 2. Coronavirus Transmission

The first outbreaks of coronavirus showed that the virus can be transmitted from person to person through the air, following the inhalation of aerosols or droplets of viral particles removed by sneezing or coughing or through direct contact with contaminated surfaces, observing that the virus resists on surfaces from a few hours to a few days, depending on their material (Figure 2).

In the opinion of the World Health Organization, the Member States of the European Union should base all their actions on the detection and isolation of symptomatic persons, but also on the conduct of epidemiological investigations that also contribute to the identification of contacts.

Preventive strategies play a major role in reducing the spread of the virus, along with isolating the community, in addition to detailed knowledge of the virus in terms of the structure and mechanism of action [14].

WHO has also considered analyzing how and to what extent the coronavirus can be transmitted from animals to humans. Thus, this finding proposes to identify those animals that are exposed to this virus in order to be used later for testing experimental anti-COVID-19 vaccines.

There have been various claims that this virus has been spread and transmitted from bats to humans because the genome of the new coronavirus bears a 96% similarity to another type of coronavirus isolated from bat species that has, thus, gone through several processes of recombination, reaching the structure of today [15].

This pandemic has affected several species of animals. The least common cases have been identified in pets (e.g., dogs and cats), which are infected with the virus transmitted by their owners. To a large extent, the human-to-animal transmission of the SARS-CoV-2 virus has been observed on mink farms in the Netherlands and Denmark, leading to the euthanasia of millions of animals. Therefore, the dynamics of this disease require a number of explanations, mainly in terms of the transmission of the virus from animals to humans and vice versa [16].

On 17 May 2021, the European Commission decided to focus their interest on the surveillance of SARS-CoV-2 infections in minks, emphasizing that the epidemiological assessment of the risk of this virus is a priority in the field of public health. Thus, an in-depth investigation was carried out by using the genome sequencing of the outbreaks in 16 mink farms and of the people who carry out their professional activity within those farms. The findings showed that the virus was transmitted by humans, later evolving and most likely reflecting large-scale circulation among minks at the beginning of the pandemic, a few weeks before it was detected. Of all the employees of those mink farms, it was found that 68% of them had symptoms of SARS-CoV-2 infection [17].

An important aspect in the transmission of the virus is related to the time of keeping the virus active on different surfaces, either porous or non-porous. The maintenance of the virus on different surfaces depends on a number of factors, such as biological factors: type of virus and its structure, and environmental factors: temperature, humidity, light, and the physicochemical properties of surfaces, especially porosity and pH [18].

Hard surfaces, including steel and plastic, have been shown to keep the virus active for a long time, while soft and porous surfaces such as fabric, paper, and cardboard support the virus in the active state for a shorter period of time [19].

The study by Riddell et al. demonstrated that a SARS-CoV-2 viral load (final concentration of 3.38 × 10^5^/10 µL inoculum), equivalent to that normally transmitted from an infected patient, remains active for up to approximately 28 days on non-porous surfaces such as glass, stainless steel, paper, and vinyl, at a temperature of 20 °C and relative humidity of 50%. Under the same conditions on the porous materials, more precisely on cotton cloth, the recovery of the infectious virus was lower, producing an absorption effect, not being detected after 14 days from the inoculation. At 30 °C, the active virus remained on non-porous surfaces (stainless steel, glass) for up to 7 days and on cotton for only 3 days. At a higher temperature (40 °C), the recovery of the virus was much lower compared to the tests performed at 20 °C and 30 °C. Thus, the active virus could not be recovered from the non-porous surfaces after 48 h and from the porous ones (cotton) after 24 h [20].

Temperature and humidity are key factors in the survival of the virus; an increase in them causes harmful effects on survival. In the study conducted by Chen et al., relative humidity higher than 65% explains the shorter maintenance of the virus on different surfaces (paper, wood, cotton, glass, plastic, and stainless steel) [21].

In addition, it has been observed that exposure to simulated sunlight quickly inactivates the SARS-CoV-2 virus from various surfaces or in the air [22,23].

Knowing the exact probability that a person will be infected after exposure to different contaminated surfaces is a difficult and complex operation as there may be cases in which certain persons who come into contact with these surfaces do not become infected [24].

Possible mutations in the SARS-CoV-2 virus are accelerated by its rapid transmission over a long period of time. The emergence of new variants cannot be controlled only by adopting strict health measures at the local level, this being a phenomenon encountered globally. A massive increase in the number of infections also increases the number of mutations, with the possibility of new variants with a higher degree of infectivity [25].

The basic reproduction number (R_0_) appreciates the number of new infections given by a single infected person. This parameter R_0_ refers to the early stages of a pandemic, when no sanitary measures have been taken; there is no immunity among the population, and all people are susceptible to infection. In the absence of physical distancing measures, R_0_ shows that each infection causes another 2–4 infections. As a result of social distancing and other public health measures, in September 2021, the reproductive number decreased <1 in several regions, but at the beginning of December in South Africa, the value of R increased over 2 due to the new variant. It is currently estimated that the Omicron strain will infect up to six times more people than the Delta variant [24,26].

In their paper, Liu and Rocklöv reported an R_0_ of 5.08 for the Delta variant compared to 2.79 for the original base strain [27].

## 3. Clinical Symptoms

Due to the fact that the intra-community transmission of the virus is happening faster and faster, COVID-19 is considered an infectious disease that has become responsible for the death of over 2 million people worldwide. SARS-CoV-2 infection includes non-specific clinical manifestations common with other viral diseases [28].

The World Health Organization has stated that the average incubation period for the SARS-CoV-2 virus is 6 days, but it can be influenced by various factors, especially host factors, and thus can vary from one day to two weeks [29].

The symptoms observed in infected patients were classified into three main categories: respiratory symptoms, the most common being: cough, dyspnea, sore throat, chest pain; non-specific neurological symptoms: fatigue, headache, anorexia, and muscle pain and also digestive symptoms: nausea, vomiting, diarrhea and gastrointestinal pain (Figure 3) [30,31].

In addition, the mutations in the SARS-CoV-2 virus, which led to the appearance of several strains, highlighted the different dominant symptoms for each variant.

In terms of symptoms, the Alpha strain does not differ much from the initial version of COVID-19 disease, the main symptoms being: fever, persistent cough, anosmia, and dysgeusia. Additionally, with the rapid spread of the Alpha variant, additional symptoms were revealed, such as: chills, muscle aches, headaches, and loss of appetite [32]. For the Beta variant, there is no clear finding that the symptoms of this strain are different from those of SARS-CoV-2 variants previously discovered. In a prospective cohort study, it was found that myalgia and coryza are the most common symptoms in Gamma strain cases, and a decrease in the frequency of dysgeusia and anosmia was observed [33]. According to the Zoe COVID Symptom Study, the most commonly reported symptoms of the Delta strain were: headache, sore throat, runny nose, cough, and sometimes fever, chills, and shortness of breath [34]. The new Omicron variant, which has a higher rate of transmissibility, has milder symptoms than the existing strains. The most common symptom is severe fatigue with muscle and headaches; most infected people do not require hospitalization. In addition, the loss of smell and taste is not among the main symptoms reported by patients [35].

## 4. Complications

Most patients develop only mild or moderate forms of the disease, with symptoms common to a viral infection such as cough, fatigue, fever, anorexia, loss of taste, and smell. Severe forms of the disease begin with common symptoms, and, from the seventh day, the patient’s condition may deteriorate, requiring hospitalization. At this stage, the appearance of pneumonia with hypoxemia and increases in liver enzymes and creatine have been observed and can lead to serious complications.

The most common complication is respiratory failure, where it is necessary to resort to respiratory support.

Additionally, cardiac complications have an increased incidence in patients admitted to intensive care, arrhythmias and hypoxemic cardiomyopathy being the most presented [15].

Various coagulation disorders [36,37,38] and acute inflammatory conditions sustained by the presence of inflammatory markers in a large percentage have been diagnosed [39].

In addition, the high mortality rate was associated with the insufficiency of various organs [15].

Knowing the clinical manifestations, it is necessary to approach the therapy depending on the severity of the disease.

Of the 30 known coronaviruses types that infect humans and animals, only α- and β-models are zoonotic in origin and are strongly pathogenic to humans. Human α-coronaviruses, including NL63, 229E, OC43, and HKU1, generally slightly affect the upper respiratory tractus in the case of immunocompetent patients, with common symptoms of a classic cold: fever, headache, and cough. Human β-coronaviruses include SARS-CoV and MERS-CoV, with mild and no clear sign symptoms at the primary stage; then, it can quickly advance to severe pneumonia, dyspnea, and even death, with morality rates of ~10% in the case of immunocompetent patients and ~35% in the case of immunocompromised patients [40].

## 5. COVID-19 Treatment

Detailed research is currently being conducted to identify drugs with a role in treating COVID-19 infection as well as other coronaviruses. A general description of therapeutic agents that have shown a beneficial effect in the treatment of viral infection with COVID-19 will be provided below [41].

To date, there is no clear treatment for COVID-19 and there are no drugs approved only for the treatment of COVID-19, remdesivir being the only substance that has been approved for use by the Food and Drug Administration (FDA) in the treatment of coronavirus disease. The care of patients infected with COVID-19 includes several steps such as: early diagnosis, isolation of the patient, application of various methods of protection to prevent the spread of infection, and supportive treatment [1].

Different drug classes have been investigated in different stages of COVID-19 disease; the main classes observed as having benefits are: antiviral drugs (remdesivir, ribavirin), antibodies (convalescent plasma, immunoglobulins), immunomodulatory drugs (tocilizumab, siltuximab), anti-inflammatory drugs (dexamethasone), as well as antimalarial drugs (chloroquine/hydroxychloroquine).

### 5.1. Antiviral Medication

Antiviral medication given to patients with COVID-19 was initially approved for the treatment of other infections such as influenza, human immunodeficiency virus (HIV) infection, or Ebola. Thus, the goal of medical researchers has been to test the existing compounds that are known to be effective in viral infections to accelerate the development of an effective treatment in the fight against COVID-19 disease [42].

#### 5.1.1. Remdesivir

In October 2020, remdesivir was approved by the FDA for the treatment of COVID-19 infection, and it was, the first drug approved for the treatment of viral infection in adults and children (over 12 years of age and weighing at least 14 kg) who require hospitalization [43].

Remdesivir (Figure 4) is an antiviral drug that disrupts viral replication by inhibiting the ribonucleic acid (RNA) polymerase; in addition, it is an analog of adenosine, a prodrug, which has a broad spectrum of activity against several families of viruses, such as *Pneumoviridae*, *Filoviridae*, and *Paramyxoviridae* [44,45].

Remdesivir is a well-tolerated therapeutic agent, the main side effects being nausea, decreased blood pressure, increased liver enzymes (aminotransferase levels), and respiratory failure. Following the administration of remdesivir, it was concluded that it should not be used in people with severe liver disease and kidney failure [46,47].

Remdesivir is an antiviral drug in the class of nucleotide analogs, developed to treat infections caused by RNA viruses such as Ebola, Nipah, and MERS. Today, it is being extensively studied for use in the treatment of COVID-19 infection [48,49].

The SARS-CoV virus uses angiotensin-converting enzyme 2 (ACE 2) as a gateway to infect the cell via the spike viral protein and entry receptor-mediated endocytosis. The mechanism of action of remdesivir is based on the resemblance of its active metabolite to the substrate required for ribonucleic acid synthesis. First, remdesivir is metabolized in the host cell to a metabolite of alanine, which then undergoes several transformations; a monophosphate derivative is obtained and, finally, the nucleoside analog triphosphate. In the cell infected with SARS-CoV-2, the active metabolite interferes with the action of the RNA polymerase dependent on viral RNA and slows down the activity of exoribonuclease, which corrects newly synthesized RNA. The attachment of the active derivative and its incorporation into the developing RNA strand stops strand elongation and inhibits RNA synthesis (Figure 5) [50,51,52,53].

Remdesivir and the antimalarial drug chloroquine have shown a beneficial effect against COVID-19 in vitro. The effect of the two drugs on Vero E6 cells infected with nCoV-2019 BetaCoV/Wuhan/WIV04/20192 was analyzed, looking at the effects on virus yield, cytotoxicity, and 2019-nCoV infection rates. Following the results, remdesivir showed a 50% cytotoxic concentration of >100 μM, a half-maximal effective concentration of 0.77 μM, and a selectivity index of >129.87 [54].

Additionally, in a mouse SARS-CoV model, the drug decreased viral load and lung disease [42]. Numerous clinical trials have shown a beneficial effect on COVID-19 disease.

Gilead Sciences has conducted a phase III clinical trial with the antiviral drug remdesivir to demonstrate the efficacy and safety of the molecule in use as a treatment for COVID-19. It was observed that the therapeutic agent produced a reduction in the mortality rate after 14 days of treatment and an improvement in the condition in 64% of cases. Thus, following the results obtained, the FDA authorized the use of remdesivir as a therapeutic agent against COVID-19 disease [55].

A double-blind, placebo-controlled clinical trial found that the remdesivir group had a recovery time of 11 days compared to the 15-day placebo group. In addition, of the 541 patients who received remdesivir, only 114 reported serious adverse events [56].

Intravenous administration of remdesivir for 10 days to 53 patients confirmed with COVID-19 led to an improvement in the health of 36 of them, following administration of 100 mg of the substance for 9 days, with a dose administered on the first day of 200 mg [57].

#### 5.1.2. Lopinavir/Ritonavir

The antiviral combination lopinavir/ritonavir represents two substances in the class of protease inhibitors, initially approved in the treatment of HIV. Lopinavir is responsible for the antiviral activity of lopinavir/ritonavir. Lopinavir is an inhibitor of HIV-1 and HIV-2 proteases. The inhibition of the HIV protease prevents the cleavage of gag-pol polyprotein and results in the production of immature, non-infectious viral particles.

The co-administration of the two drugs is beneficial; ritonavir (Figure 6a) inhibits the metabolism of the Cytochrome P450 3A4 (CYP3A4) isoenzyme and thus increases the plasma concentration of lopinavir (Figure 6b) [58].

The most common adverse reactions reported with lopinavir/ritonavir in clinical trials were diarrhea, nausea, vomiting, hypertriglyceridemia, and hypercholesterolemia. Diarrhea, nausea, and vomiting may occur at the beginning of the treatment, while hypertriglyceridemia and hypercholesterolemia may occur later in the course of the treatment.

With the rapid spread of the virus, this combination has been proposed as an antiviral treatment for COVID-19.

The doses of lopinavir/ritonavir given to treat COVID-19 are 400/100 mg twice a day for two weeks. In a study by Cao, the combination of antiviral lopinavir/ritonavir reduced intensive care hospitalization by 5 days but did not show significant differences from standard medication for improvement in clinical manifestations, viral clearance, or mortality rate at 28 days [59].

A randomized, single-blind, controlled study in 34 patients with mild to moderate COVID-19 disease found that there were no differences in the negativity of the Reverse transcription polymerase chain reaction (RT-PCR) test in patients receiving the viral combination compared to the control group. In addition, there was no evidence of cough relief or improvement in computed tomography (CT) between the two groups [60].

Unfortunately, the studies performed do not seem to be sufficient due to the small number of subjects studied, the late administration of treatment, suboptimal doses of the antiviral combination, and the absence of co-administration with other antivirals, e.g., ribavirin.

#### 5.1.3. Ribavirin

Ribavirin (Figure 7) is a guanosine nucleoside analog, a broad-spectrum antiviral that acts against RNA viruses, with a primary indication for hepatitis C and viral hemorrhagic fevers. At the same time, it may be efficient in the early stages of viral hemorrhagic fevers, including Lassa fever, Venezuelan hemorrhagic fever, Crimean-Congo hemorrhagic fever, and Hantavirus infection. More recently, it has been tested against SARS-CoV [61]. Ribavirin also possesses an immunomodulatory action of the host to the virus by shifting a Th2 response in favor of a Th1 phenotype. Th2 response and production of type 2 cytokines such as IL-4 (interleukin-4), IL-5, and IL-10 actuate the humoral response, which enhances immunity toward the virus [62].

The most common side effects of this medication are a decrease in red blood cells and neutrophils, dizziness and anxiety, stomach pain, loss of appetite, diarrhea, and fever.

An open-label phase 2 study in 127 patients in China compared the effects of the combination of lopinavir/ritonavir—400/100 mg, ribavirin—400 mg, and interferon β-1b—8 million IU, with the effects seen when given only lopinavir/ritonavir. The group of patients who received the combination of the three drugs showed faster negativity of the RT-PCR test, shorter hospitalization, and faster relief of symptoms [63].

#### 5.1.4. Nirmatrelvil/Ritonavir

With the advent of the new Omicron variant, which has a large number of mutations and accelerated transmissibility, the scientific world has been alerted. Thus, in order to control the strain, the FDA approved on 22 December 2021 the oral antiviral Paxlovid™ (nirmatrelvil + ritonavir), (Pfizer Inc., New York, NY, USA), a drug that gives high hopes for recovery in COVID disease [64]. Paxlovid™ has been approved for use in the treatment of mild to moderate forms of COVID disease in both adults and adolescents and children >12 years of age who weigh at least 40 kg. Paxlovid™ is a combination of nirmatrelvir, the molecule that stops viral replication by inhibiting the SARS-CoV-2 protein, and ritonavir, an anti-HIV drug that slows down the metabolism of nirmatrelvir through inhibiting cytochrome P450 enzymes. Nirmatrelvir (PF-07321332—Figure 8), the SARS-CoV-2 main protease (Mpro) inhibitor, is a novel molecule that combines the effects of both the SARS-CoV-2-3CL protease inhibitor (PF-07304814) and boceprevir (protease inhibitor used to treat hepatitis caused by hepatitis C virus genotype 1) [65,66,67].

Nirmatrelvir is a broad-spectrum antiviral agent used in vitro to treat several coronavirus infections such as SARS-CoV-2 and MERS-CoV. The group led by Owen et al. reported that nirmatrelvir has a strong anti-SARS-CoV-2 main protease action in Vero E6 cells without notable cytotoxicity. This molecule has also been shown to improve the anti-SARS-CoV-2 action in a mouse-adapted SARS-CoV-2 MA10 model, improving multifocal lung damage and reducing the viral load in the animal’s lungs, with a dose-dependent effect [68].

The first clinical data that revealed the effect of Paxlovid™ in the treatment of SARS-CoV-2 infection in adults were reported in an interim analysis of the phase 2/3 EPIC-HR (Evaluation of Protease Inhibition for COVID-19 in High-Risk Patients) randomized, double-blind, placebo-controlled trial. During the 28-day study, of the 1039 non-hospitalized patients who received the newly approved drug, only 0.8% were hospitalized or died compared to 7% for placebo [69].

Side effects that may occur after taking this combination of drugs are: high blood pressure, changes in taste, muscle aches, and diarrhea. In addition, ritonavir may damage the liver, so Paxlovid™ should be used with caution in patients with liver disease [64].

Existing data support the efficacy of the new drug as a promising therapeutic agent in COVID-19 disease, but more experiments are needed to demonstrate its safety.

### 5.2. Antimalarial Drugs

#### Chloroquine and Hydroxychloroquine

Chloroquine (Figure 9a) is a drug that belongs to the class of antimalarials, with a basic structure of 4-aminoquinoline [70]. This drug inhibits the action of heme polymerase, which causes the buildup of toxic heme in Plasmodium species [71].

Besides its wide use in the fight against malaria, chloroquine has a therapeutic effect in the treatment of HIV infection and in rheumatoid arthritis, having anti-inflammatory and immunomodulatory activity [72]. The most frequent adverse effects are headache, drowsiness, visual disturbances, nausea, vomiting, and hypokalemia.

Hydroxychloroquine (Figure 9b) is a metabolite of chloroquine with fewer side effects, also used in malaria and in the treatment of rheumatic pathologies [73]. The substance accumulates in the lysosomes of the malaria parasite and in human organelles, raising their pH, which inhibits antigen processing, preventing the α and β chains of the major histocompatibility complex class II from dimerizing, inhibiting antigen presentation of the cell, and reducing the inflammatory response [74].

The known action of the two substances to increase endosomal pH supports their antiviral activity by inhibiting viral replication. In addition, studies have shown that these therapeutic agents inhibit the glycosylation of the angiotensin-converting enzyme 2, located on the cell membranes of the lungs, kidneys, and heart, an enzyme that is involved in the cellular penetration mechanism of the new virus (SARS-CoV-2) [54,75].

An important point to note is that hydroxychloroquine is a human Toll-like receptor (TLR) blocker and can inhibit endosomal TLR3, -7, -8, and -9 signaling, thus controlling inflammation in COVID-19 disease and ameliorating the negative effects of SARS-CoV-2 infection [76,77].

Thus, chloroquine and hydroxychloroquine have received special attention in COVID-19 disease, with numerous studies that have highlighted their effect in viral infection with SARS-CoV-2.

Chloroquine and hydroxychloroquine have demonstrated antiviral action on SARS-CoV-2-infected Vero cells. It has been found that hydroxychloroquine is more active than chloroquine, having a half-maximal effective concentration (EC50) of 0.72 μM, with chloroquine having an EC50 of 5.47 μM; hydroxychloroquine is known to be a safer and more tolerable molecule [78].

In a non-randomized observational study led by Gautret et al., the effect of azithromycin co-administered with hydroxychloroquine was investigated in a group of 80 patients with COVID-19 disease. According to the results, 83% of subjects had a negative nasopharyngeal test 7 days after starting treatment [79].

In another non-randomized study, 20 patients with severe COVID-19 were given hydroxychloroquine co-administered with azithromycin or not. Following treatment, it was observed that hydroxychloroquine decreased the viral load, and this effect was intensified with the administration of azithromycin [80].

In an observational study, the clinical outcomes of patients confirmed with COVID-19 who received or did not receive hydroxychloroquine were compared. No major differences were found between the two groups; the administration of hydroxychloroquine did not reduce the risk of intubation or death [81].

A multicenter retrospective study demonstrated that the administration of hydroxychloroquine with or without azithromycin to 807 patients diagnosed with COVID-19 was not effective or even had a potentially harmful effect. The two substances did not reduce mortality or the need to use mechanical ventilation. The group receiving only hydroxychloroquine had a higher risk of death [82].

To date, the results of studies on the effect of chloroquine and hydroxychloroquine in COVID-19 disease are contradictory. Thus, more randomized clinical trials are needed to understand whether the benefits of these drugs outweigh the risks.

### 5.3. Corticosteroids

Corticosteroids are therapeutic agents in the class of anti-inflammatory drugs; they are medications used in the treatment of various pathologies, from respiratory and allergic diseases to autoimmune diseases and cancer. Besides the numerous therapeutic effects, corticosteroids also have various adverse effects, such as: hyperglycemia, hypertension, bone damage, an increased risk of infections, and the development of obesity [83].

The immunomodulatory action of corticosteroids is beneficial in viral infections, including SARS-CoV-2 infection, reducing the inflammatory response. However, special attention should be paid to the dose of corticosteroids and to the period of administration due to the fact that these drugs can affect the immune system, increasing the viral load. Therefore, they are recommended for patients who have a more severe form of the disease, who are dealing with a cytokine storm [41].

With the outbreak of the pandemic, the treatment with corticosteroids in patients with COVID-19 began. In Wuhan, following a retrospective cohort study of 201 confirmed COVID-19 patients with acute respiratory distress syndrome, it was observed that patients who were treated with methylprednisolone had a lower mortality rate compared to patients who were not given corticosteroids [84].

Another clinical study was performed in a larger number of patients, which looked at the effect of dexamethasone in patients with COVID-19 compared to patients receiving standard care. It has been reported that taking 6 mg daily of dexamethasone for 10 days reduced the death rate in patients who required respiratory support and who had symptoms for more than a week [85].

According to another study, dexamethasone reduced mortality by one-third in confirmed COVID-19 patients compared to patients who received standard care [86].

Currently, the use of glucocorticoids in the treatment of SARS-CoV-2 infection is controversial because of its immunosuppressive effects. Thus, the benefits and risks must be clearly investigated before their clinical use.

### 5.4. Immunomodulatory Drugs

#### Interleukin-6 Inhibitors

Tocilizumab is a monoclonal antibody, an (interleukin-6) IL-6 receptor antagonist. It is used in the treatment of inflammatory diseases such as rheumatoid arthritis or cytokine release syndrome. Interleukin-6 levels correlate with COVID-19 severity, so the investigation of IL-6 inhibitors is strongly justified [87]. The most common side effects of this molecule are: increased cholesterol levels, increased alanine aminotransferase and aspartate aminotransferase, and allergic reactions [88].

In a multicenter study of 21 patients with severe COVID-19, it was observed that tocilizumab normalizes temperature, reduces oxygen demand, and improves CT imaging [89].

Another representative of the IL-6 inhibitors, siltuximab, initially approved for the treatment of Castleman’s disease (rare lymphoproliferative disorder) [90], has been studied for its efficacy in patients with COVID-19. Gritti et al. have shown that taking siltuximab at a dose of 900 mg in patients with COVID-19 and acute respiratory distress syndrome reduces the serum level of C-reactive protein and improves clinical manifestations, with no deterioration in the general condition [91].

### 5.5. Antibodies

#### 5.5.1. Convalescent Plasma Therapy

Convalescent plasma is obtained from patients who have been cured of COVID-19 disease and contains neutralized antibodies to the virus that are then given to patients infected with SARS-CoV-2 to help the immune system and increase the patient’s immune response to the virus [1].

In addition, this therapy blocks the infection and improves the clearance of infected cells and the condition of severely affected patients [92,93].

Convalescent plasma treatment is not a new concept; this type of therapy has shown good results in various viral diseases, such as Ebola or H5N1 influenza [94,95].

Following a study performed on 25 patients diagnosed with COVID-19, the effect of convalescent plasma in this pathology was followed; hence, for 19 of the patients, a clinical improvement of at least 1 point was observed on the ordinal route of the WHO, with the help of which the severity of the disease is calculated [96].

In another study of 10 patients with COVID-19, it was shown that once convalescent plasma was administered, it could maintain or even increase the antibody titers received. In addition, clinical symptoms improved, decreasing viral load. Following this study, it was concluded that convalescent plasma therapy can be well tolerated by patients, with no side effects observed [97].

Another analysis of 5000 patients with severe COVID-19 who received convalescent plasma showed that less than 1% of patients had a severe side effect within the first 4 h after administration. The observed serious side effects, which may be related to plasma administration, were: severe allergic reactions (3 cases), acute lung damage (11 cases), circulatory overload (7 cases), and death (4 cases) [98].

A randomized study of 103 patients with severe COVID-19 did not achieve a statistically significant improvement in the time to symptom relief within 28 days between those receiving standard treatment (43.1%) and those who were additionally given convalescent plasma (51.9%). The clinical trial was stopped too early due to the decrease in the number of enrolled patients, the interpretation of the data being limited, and it is not possible to specify whether there is a clinically important difference [99].

However, existing studies need to be supplemented by new well-controlled clinical trials to establish a clear protocol for convalescent plasma in patients diagnosed with COVID-19.

#### 5.5.2. Special Monoclonal Therapy

Treating SARS-CoV-2 infection with antiviral medication alone may not be enough to control the infection and associated pathologies, so treatment with monoclonal antibodies may be an effective strategy to combat COVID-19 disease.

A therapeutic agent in the monoclonal antibody class was developed by Lilly, which is bamlanivimab (LY-CoV555), obtained from the convalescent plasma of patients infected with COVID-19 [100].

In a phase 2 clinical study performed on outpatients, it was observed that on day 11, bamlanivimab reduced the viral load and severity of symptoms compared to the placebo group.

In a retrospective case–control study realized by Rebecca Kumar et al. on 403 subjects, 218 patients received bamlanivimab. Thus, the 30-day hospitalization rate was 7.3% of those patients compared to 20.0% of patients who did not receive bamlanivimab [101].

Another research pointed out, however, that this monoclonal antibody is not very sensitive to the Delta mutation (besides D614G, the D950N mutation mapped to the trimer interface), one of the newest mutations encountered [102].

To increase the viral load reduction, bamlanivimab was combined with etesevimab, the association showing a statistically significant reduction in SARS-CoV-2 viral load at day 11, more effective than in the case of the individual administration of each [103].

In addition, Regeneron designed a mixture of two monoclonal antibodies (casirivimab and imdevimab), called REGN-COV2, from humanized VI mice and patients cured of COVID-19. The two recombinant human IgG1 monoclonal antibodies target the receptor-binding domain of the spike protein of SARS-CoV-2, thus neutralizing viral entry into human cells via the angiotensin-converting enzyme 2 receptor. This drug was able to annihilate the virus and prevent it from binding to the host cell, preventing infection [104].

REGN-COV2 is being investigated in several clinical trials. Following phase 2/3 studies performed on both hospitalized and non-hospitalized patients, a reduction in the clinical symptoms of those non-hospitalized was observed [105,106].

A retrospective study analyzed the difference observed in the patients who were administered bamlanivimab or casirivimab + imdevimab. Results showed that the combination of the two antibodies reduced the hospitalization rate (2.83%) better than bamlanivimab (4.34%) [107].

The effectiveness of the medication in immunocompromised patients following kidney transplantation was analyzed; the antibodies were well tolerated and significantly reduced the hospitalization rate [108].

In the case of pregnant women, the medication has proven to be just as safe and effective [109].

As we can see, there are studies that have shown the effectiveness of monoclonal antibodies on the SARS-CoV-2 virus. However, the new experiments performed on the new Omicron variant show a decrease in the activity of monoclonal antibodies, referring to both FDA-approved antibodies, such as those from Eli Lilly and Regeneron, those recommended by European Medicines Agency (EMA) from Celltrion, and those under development.

The multitude of mutations in strain B.1.1.529 leads to poor efficacy or ineffectiveness of this therapy. The rapidly occurring SARS-CoV-2 variants endanger antibody therapy [110].

Due to the fact that the newly discovered variant has a high transmission rate and that obtaining reliable results from experimental research laboratories can take up to a few weeks, in silico analysis has begun, this being an urgent situation.

One such study was conducted by Chen et al., which looked at how the changes in the S protein receptor-binding domain on the new strain affect the degree of viral infectivity and whether existing antibody therapy is effective on this variant of the virus.

This experiment, which used an artificial intelligence model, showed that the B.1.1.529 variant can be up to 10 times more contagious than the original SARS-CoV-2 variant and twice as contagious as the Delta strain, the strain that has caused the highest morbidity.

In the same study, the combination of Eli Lilly antibodies (bamlanivimab and etesevimab) was shown to have very low efficacy on the Omicron strain. In addition, changes in the spike protein impair the efficacy of other monoclonal antibodies, Regeneron antibodies (casirivimab and imdevimab), Celltrion monoclonal antibodies (regdanvimab—CT-P59 and CT-P63), and those in clinical trials at Rockefeller University [13].

The scientific group led by VanBlargan evaluated the action of various antibodies used in monotherapy or combination therapy on the isolated strain of B.1.1.529.

Their results showed that monoclonal antibody mixtures such as Regeneron (REGN10933 and REGN10987), Lilly (LY-CoV555 and LV-CoV016), and Celltrion (CT-P59) almost completely lost their neutralizing activity against strain B.1.1.529 in studies performed on Vero-TMPRSS2 and Vero-hACE2-TMPRSS2 cells, while combinations of AstraZeneca antibodies (VOC-2196 and VOC-2130) or Vir Biotechnology (S309) continued to show inhibitory action [111].

In vitro experiments have shown that certain monoclonal antibodies used in therapy may be effective on the new Omicron strain, but detailed in vivo studies are needed to validate and support the results obtained on the cells.

Administration of monoclonal antibodies could be an effective option in treating viral diseases and even in treating COVID-19 infection. However, studies should be continued on a large number of patients to confirm the results. The effectiveness of monoclonal antibodies depends a lot on the structure of the isolated strain, and their poor efficacy is observed on the Omicron variant, which has various mutations compared to the basic strain.

In studying the treatment for COVID-19 disease, the essential role of cholesterol in SARS-CoV-2 replication was observed, a direction that needs to be further explored. Cholesterol is a molecule with implications for various cellular processes, including the penetration of the SARS-CoV-2 virus into the host cell. Cholesterol-rich lipid rafts represent a surface through which viruses, through endocytosis, enter the cell and are also essential for the interaction between protein spike and the angiotensin converting enzyme 2 receptor [112]. Significantly, high-density lipoproteins (HDL) can facilitate virus entry into the host cell via the SR-B1 receptor, capture viruses, or mediate their entry. Thus, people with lipid profile disorders may be more prone to SARS-CoV-2 infection than the healthy population. Metabolic lipid pathways and membrane structure should be studied to inhibit the life cycle of the virus as a basis for antiviral medication. A drug used to treat patients with COVID-19, hydroxychloroquine, may disrupt cholesterol-rich lipids, leading to the regulation of key immune signaling entities caused by SARS-CoV-2 infection. Therefore, the role of cholesterol in coronavirus replication may guide new therapies in the fight against the SARS-CoV-2 virus [113,114].

It is important to note that these drug classes discussed in the review, which have been shown to be effective in COVID-19 disease, were initially introduced into therapy for other pathologies. Table 1 illustrates the main representatives, together with the diseases in which it is administered and the mechanism of action.

## 6. Implications of Malpractice in the Context of the SARS-CoV-2 Crisis

Currently, the progress in medicine is spectacular. In the context of this pandemic, the efforts of the European international authorities for the treatment and development of vaccines for all Member States of the European Union should be noted.

In order to combat this virus, the World Health Organization has drawn attention to the fact that the emergence of a state of health emergency has mobilized all states around the world to take all beneficial measures to help stop this pandemic.

Internationally, there are a number of normative acts issued before the emergence of the pandemic, the applicability of which is now proving beneficial.

One of the regulations under which the international community sought to implement the principles applicable in the process of developing the SARS-CoV-2 virus vaccine was the Astana Declaration, adopted following the Global Conference on Primary Health Care. The Declaration summarizes the main directions of action, which incorporates primary health care and issues related to vaccines and vaccination.

At the European level, the issue of developing a vaccine as an urgent remedy for the COVID-19 pandemic has been raised. In this context, on 17 June 2020, the European Union Strategy for Vaccines against COVID-19 was developed. The document shall take into account the risks and costs of ensuring the quality, safety, and efficacy of vaccines.

The period in which human society is today is completely different from that before the pandemic, and what is happening in the world today is the subject of medical research, analysis, and reflection for all scientists. Consequently, ensuring the appropriate regulatory framework in each country is an urgent need, the measures imposed and their strict observance being the main means by which the pandemic can be stopped. In addition to the various therapeutic problems that this disease raises, it can even generate pandemic risk situations for the doctor, engage his legal liability, and lead to various sanctions.

Within the meaning of article 653 par. (2) of the content of Law no. 95/2006 on health care reform, medical staff may be held civilly liable for damages they cause to patients due to error, negligence, recklessness, or insufficient medical knowledge during the provision of medical care in the prevention, diagnosis, or treatment of disease.

In the specialized doctrine [130], a distinction was made between “medical mistake” and “medical error” in the context of the coronavirus pandemic, and it was said that medical error is a complicated evolution of the disease or symptomatology, with a tragic end. Therefore, no diligent physician could have stopped the unfortunate course of the disease.

The medical mistake, on the other hand, is characterized by the fault of the medical staff in the exercise of the profession, manifested in the form of recklessness or negligence.

Negligence in the present case is reported as an omission to act, which may be limited to haste, superficiality, or improper performance of duties [131].

Our assertions in this article cannot and do not aim to be unique solutions to all coronavirus problems. They represent open doors for new forensic research that will follow in the future.

## 7. Conclusions

The SARS-CoV-2 pandemic has become a global medical priority. Transmission from one person to another has led to the accelerated spread of the virus in the community, making it increasingly difficult to control. Although COVID-19 appears to be a simple viral infection, it can still be fatal. With the onset of the pandemic, scientists around the world have begun to work together to protect the population from this “invisible enemy.” Considerable efforts have been made to understand the pathophysiology of COVID-19, leading to scientific advances in anti-COVID-19 treatment.

For these reasons, the WHO recommends the application of supportive treatments and careful management of complications.

The exact pathogenesis of COVID-19 disease remains unclear but has been found to usually involve a disordered hyperinflammatory response after viral infection. Moreover, in addition to the host response, variations in the viral strain can contribute to the severity of the disease and its spread.

At this time, little is known about B.1.1.529 infectivity and antibody resistance. Thus, more and more studies on mutations in Omicron strain are needed to understand the process and stop accelerated transmission.

There are currently other drugs in various therapeutic classes that are being studied for their effectiveness in COVID-19 disease, but clear evidence is expected for their therapeutic use.

## Figures and Tables

**Figure 1 medicina-58-00261-f001:**
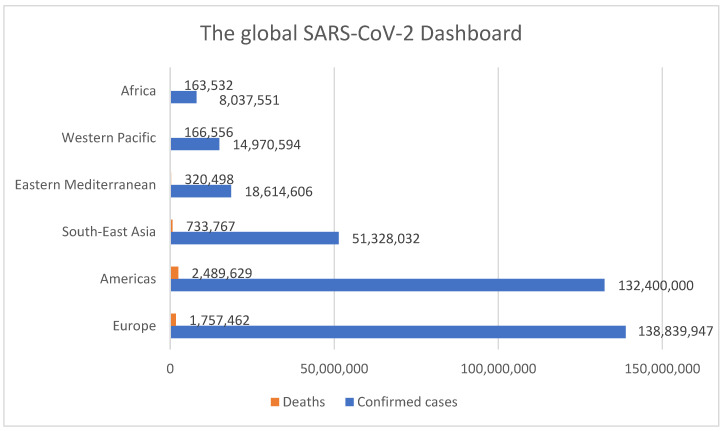
The global SARS-CoV-2 dashboard organized by WHO Regions. Data source: WHO.

**Figure 2 medicina-58-00261-f002:**
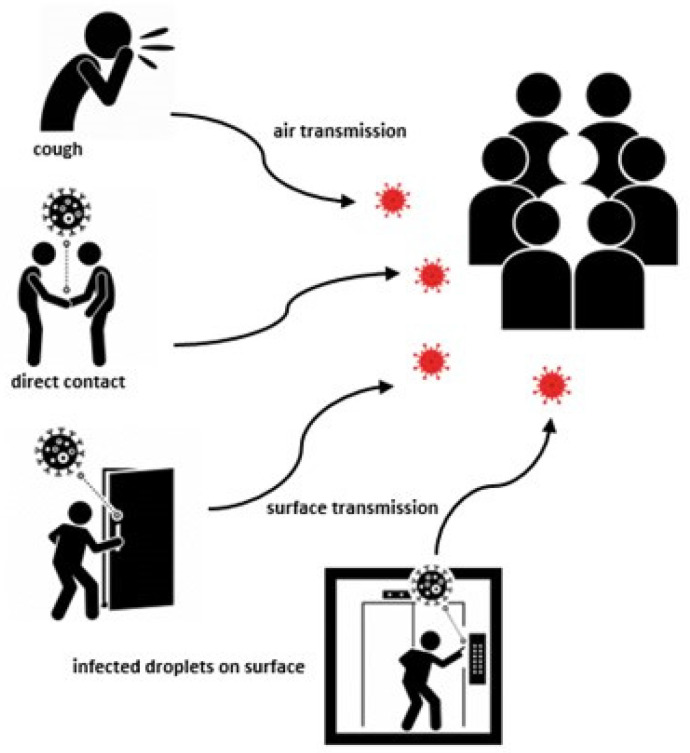
SARS-CoV-2 virus transmission pathways.

**Figure 3 medicina-58-00261-f003:**
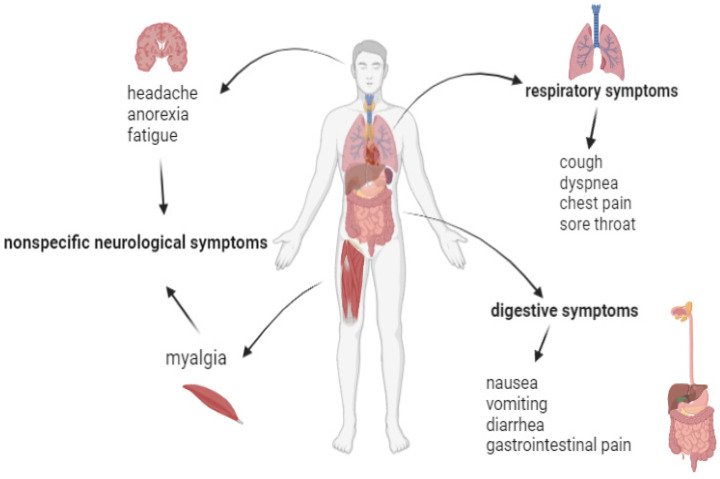
Main clinical manifestations in COVID-19 disease.

**Figure 4 medicina-58-00261-f004:**
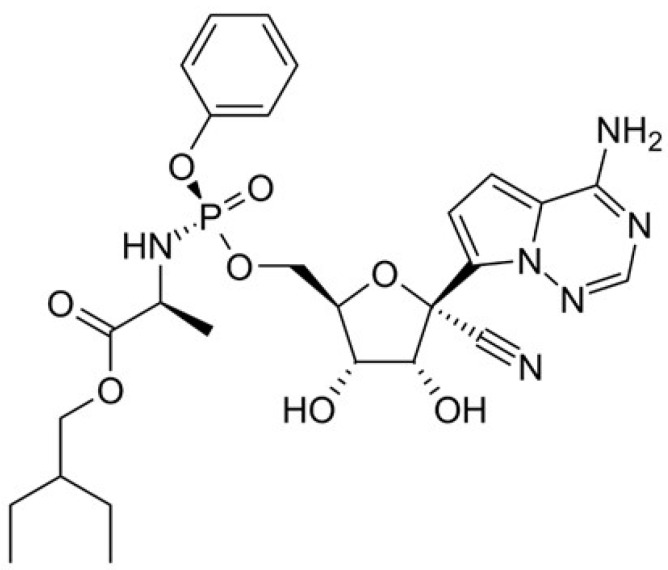
Chemical structure of remdesivir.

**Figure 5 medicina-58-00261-f005:**
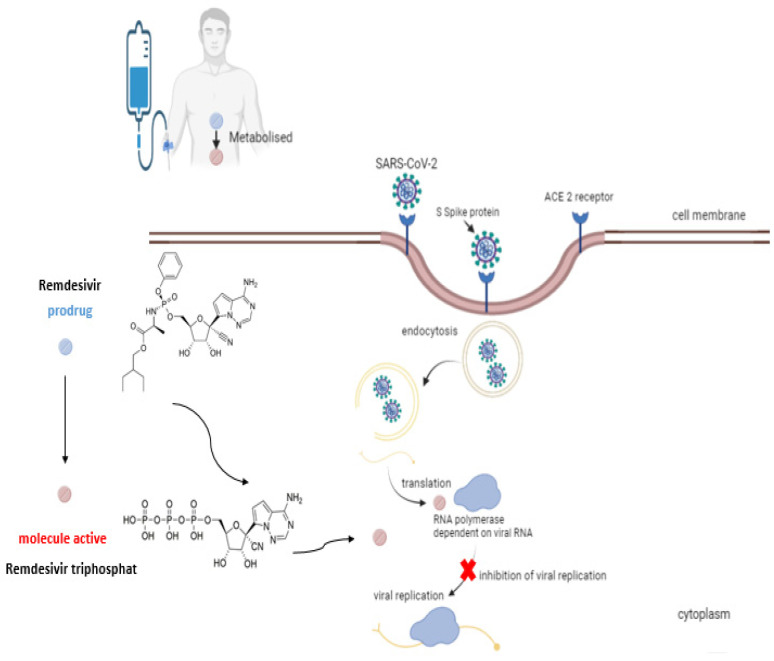
Mechanism of activity of remdesivir.

**Figure 6 medicina-58-00261-f006:**
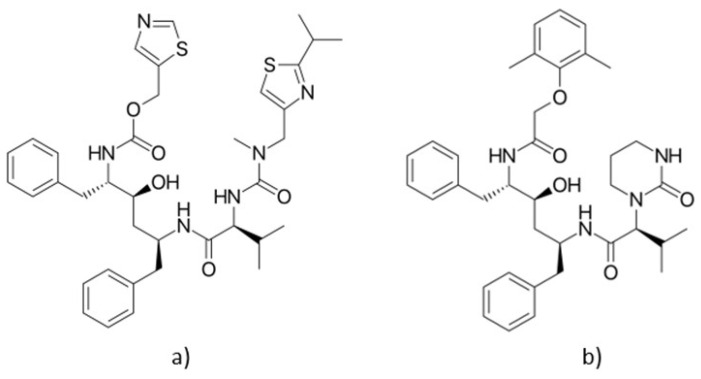
Chemical structure of (**a**) ritonavir and (**b**) lopinavir.

**Figure 7 medicina-58-00261-f007:**
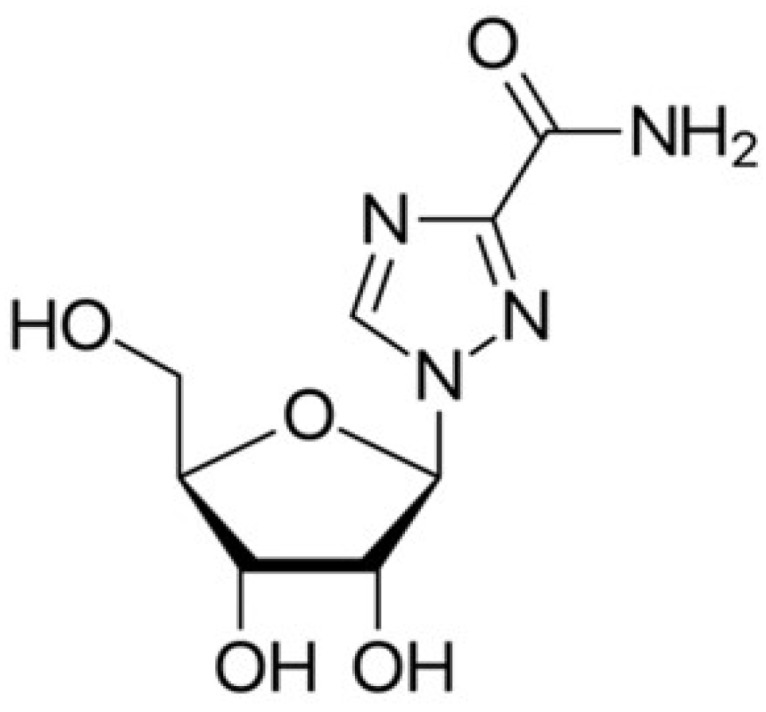
Chemical structure of ribavirin.

**Figure 8 medicina-58-00261-f008:**
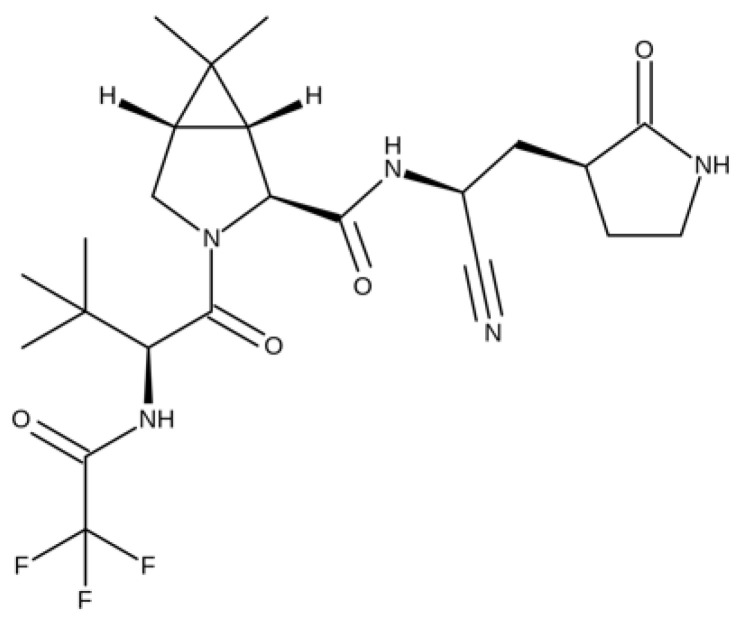
Chemical structure of nirmatrelvir.

**Figure 9 medicina-58-00261-f009:**
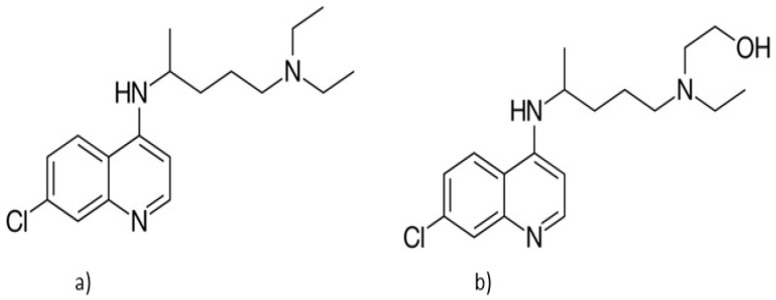
Chemical structure of (**a**) chloroquine and (**b**) hydroxychloroquine.

**Table 1 medicina-58-00261-t001:** The main pathologies for which the drugs used today in COVID-19 therapy were approved, together with the mode of action.

Drug	Disease	Mechanism of Action	Ref.
Lopinavir/Ritonavir	HIV chronic hepatitis C	HIV-1 protease inhibitor: disrupts the cleavage of protein precursors inhibition of the liver enzyme cytochrome P450 3A4	[115,116,117,118,119]
Ribavirin	chronic hepatitis C	reduces hepatitis C virus replicon colony-forming efficiency	[120,121,122]
Chloroquine Hydroxychloroquine	malaria autoimmune diseases	prevents polymerization of heme into hemozoin binds to transcriptional factors on T helper 17 cells	[123,124,125]
Dexamethasone	inflammation, asthma, allergies, drug hypersensitivity reactions, Cushing syndrome	suppresses neutrophil migration, decreases lymphocyte colony proliferation	[126,127,128]
Tocilizumab	cytokine release syndrome, giant cell arteritis rheumatoid arthritis	prevents IL-6 from binding to its interleukin-6 receptor	[88,129]

## Data Availability

The data supporting the findings of the study are available within the article.

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
