# Peer review of "The “Invisible Enemy” SARS-CoV-2: Viral Spread and Drug Treatment"

_medicina, 2022, doi:10.3390/medicina58020261_

Round 1
Reviewer 1 Report
Reviewer comments
Abstract:
The meaning of abbreviations should be written at first .
The language and grammar need to be edited
Following the documentation,..of what ?
Text :
Page 1_line 41.. Ebola and Zika virus infection in 2013, respectively 2015..this sentence is not appropriate
There is a problem in identifying the meanings of abbreviations in the whole manuscript because there are not written in the text.
Page 1_line 44. At that time a.. the authors need to revise the writing .
Line 47 .. dis-ease..why is it written like this ?
Line 57 ..the number of references sometimes come before and some times come after the fullstop
Line 58 ..some paragraphs are long ,others are short
The title give an impression that the manuscript is about Omicron ,however the authors wrote many lines about COVID 19.
Line 87..the paragraph is written without
Line 89 ..the WHO..not correct.
There are multiple punctuation errors .
Figure 2.the title is not definite only two words
Line 113 ..what tis mean by isolation of the disease
Line 115 what is meant by this study ..why there is no reference
Line 199..the aim of what
Line 362 ..the sentence is not appropriate
Author Response
Thank you very much for the time allocated for the review of the manuscript “The “invisible enemy” SARS-CoV-2: viral spread and drug treatment” as well as for your pertinent observations.
Please see below a point-by-point response to your comments:
Reviewer #1:
The meaning of abbreviations should be written at first.
Response: Thank you very much for your comment. We have adjusted it
The language and grammar need to be edited.
Response: We have taken the reviewer’s comment and we have made the change.
Following the documentation,..of what ?
Response: Thank you very much for the suggestion. We have changed it.
Page 1_line 41.. Ebola and Zika virus infection in 2013, respectively 2015..this sentence is not appropriate.
Response: Thank you very much for your comment. We have adjusted it
There is a problem in identifying the meanings of abbreviations in the whole manuscript because there are not written in the text.
Response: We have taken the reviewer’s comment and we have made the change.
Page 1_line 44. At that time a.. the authors need to revise the writing.
Response: Thank you very much for the suggestion. We have changed it.
Line 47.. dis-ease..why is it written like this ?
Response: Thank you very much for your comment. We have written the correct form.
Line 57 ..the number of references sometimes come before and sometimes come after the fullstop.
Response: Thank you very much for your valuable suggestion. We have formatted correctly.
Line 58..some paragraphs are long ,others are short
Response: Thank you very much for the suggestion. We have changed it.
The title give an impression that the manuscript is about Omicron, however the authors wrote many lines about COVID 19.
Response: Thank you very much for your comment. We have adjusted it.
There are multiple punctuation errors.
Response: Thank you very much for your comment. We have adjusted it.
Figure 2.the title is not definite only two words
Response: Thank you very much for the suggestion. We have changed it.
Line 113..what this mean by isolation of the disease
Response: We have taken the reviewer’s comment and we have made the change.
Line 115 what is meant by this study..why there is no reference
Response: Thank you very much for your comment. We have changed it.
Line 199..the aim of what
Response: Thank you very much for the suggestion. We have adjusted it.
Line 362..the sentence is not appropriate
Response: We have taken the reviewer’s comment and we have made the change.
We hope you find the revised manuscript acceptable for publication. Thank you once again for your consideration.
Best regards,
Alexandra Scurtu,
Timisoara,
31th of January, 2022

Reviewer 2 Report
The Authors have made efforts to comprehensively summarize the current state of art in the field of transmission, course and, in particular, treatment of infection caused by the SARS-CoV-2 virus. To a large extent, the authors managed to achieve this, however, in my opinion, the paper may benefit from the following corrections:
a / there are punctuation / spelling / formatting errors. Errors of this kind are placed in the following lines: 48, 201, 241, 328, 361, 385, 405 (H5N1). Nomenclature should be unified: SARS-Cov-2 or SARS-CoV-2 (correct form) / Covid-19 or COVID-19.
b / the numbering of chapters should be corrected - subchapter 5.5.1 appears twice.
c / in chapter 2 it would be appropriate to provide additional information about:
- materials / surfaces and the time for which active SARS-CoV-2 virus particles remain on them (it could be a table),
- definition and the value of the R (reproduction) coefficient ​​for each virus mutation.
d / in chapter 3 it would be worth:
- correct figure no. 3 (it is of poor quality, it is stretched),
- to provide information in the table or text about the symptoms characteristic of individual mutations of the virus, because the successive variants differed in their dominant symptoms.
e / chapter 5:
The work presents the state of art as of December 21, 2021. However, a day later - on December 22, 2021 - a very important event occurred, which means that the state of knowledge presented in the work is not up-to-date, and the sentence in line 186 is no longer true. On 22/12/2021, the FDA approved the use of the new drug - PAXLOVID (nirmatrelvir / ritonavir) - for the treatment of COVID-19. I believe that it is necessary to update the work with this information - add information about the mechanism of action of the drug, results of clinical trials, indications for use and side effects (as in other cases).
f / chapter 5.5.1: figure 5 should be enriched with a more detailed description.
Author Response
Thank you very much for the time allocated for the review of the manuscript “The “invisible enemy” SARS-CoV-2: viral spread and drug treatment” as well as for your pertinent observations.
Please see below a point-by-point response to your comments:
Reviewer #2:
a / there are punctuation / spelling / formatting errors. Errors of this kind are placed in the following lines: 48, 201, 241, 328, 361, 385, 405 (H5N1). Nomenclature should be unified: SARS-Cov-2 or SARS-CoV-2 (correct form) / Covid-19 or COVID-19.
Response: Thank you very much for your comment. We have adjusted it
b / the numbering of chapters should be corrected - subchapter 5.5.1 appears twice.
Response: Thank you very much for the comment. We have changed it.
c / in chapter 2 it would be appropriate to provide additional information about:
- materials / surfaces and the time for which active SARS-CoV-2 virus particles remain on them (it could be a table),
- definition and the value of the R (reproduction) coefficient ​​for each virus mutation
Response: We have taken the reviewer’s comment and we have made the change.
d / in chapter 3 it would be worth:
- correct figure no. 3 (it is of poor quality, it is stretched),
- to provide information in the table or text about the symptoms characteristic of individual mutations of the virus, because the successive variants differed in their dominant symptoms
Response: Thank you very much for the suggestion. We have adjusted it.
e / chapter 5:
The work presents the state of art as of December 21, 2021. However, a day later - on December 22, 2021 - a very important event occurred, which means that the state of knowledge presented in the work is not up-to-date, and the sentence in line 186 is no longer true. On 22/12/2021, the FDA approved the use of the new drug - PAXLOVID (nirmatrelvir / ritonavir) - for the treatment of COVID-19. I believe that it is necessary to update the work with this information - add information about the mechanism of action of the drug, results of clinical trials, indications for use and side effects (as in other cases).
Response: Thank you very much for your comment. We have made the additions.
We hope you find the revised manuscript acceptable for publication. Thank you once again for your consideration.
Best regards,
Alexandra Scurtu,
Timisoara,
31th of January, 2022

Reviewer 3 Report
26th January, 2022
Editor-in-Chief
Medicina
Dear Editor
I have the following comments on the article: The “invisible enemy” SARS-CoV-2: Omicron variant, viral 2 spread and drug treatment”
The review article covers aspects of the SARS-CoV-2 omicron variant. There is a significant amount of information covered in this review and it contains good illustrative figures. However, there are changes that need to be made prior to publication, including errors in syntax and error that could benefit from a copy editor.
Comments:
- Please improve the language aspects of the manuscript as there are grammatical errors and typos.
- The essential role of cholesterol in SARS-CoV-2 replication has recently been observed. How does this relate to omicron? What role does cholesterol have in entry and replication?
COVID-19 treatment (Section 5):
- Has the previous work referenced in this manuscript sufficiently separated the roles of drugs on entry, replication at the ER, and egress of SARS-CoV-2 omicron variant?
- Are there risks involved in taking medication (the drugs included in this review) for the treatment of COVID-19? Are these drugs specifically targeting viral replication or are they potentially inhibiting other stages of infection or other host pathways that the virus requires?
- The authors have described various drugs that are currently approved for the use in humans for other diseases may need to be listed in a table. In addition, these drugs may have side effects, which may need to be described and provide opinion on which of them have lesser side effects. That opinion may matter in selection of drugs to screen for anti-SARS-CoV-2 omicron variant.
By signing this letter, I approve that this article be accepted with minor modifications.
Regards,
Author Response
Thank you very much for the time allocated for the review of the manuscript “The “invisible enemy” SARS-CoV-2: viral spread and drug treatment” as well as for your pertinent observations.
Please see below a point-by-point response to your comments:
Reviewer #3:
- Please improve the language aspects of the manuscript as there are grammatical errors and typos.
Response: Thank you very much for your comment. We have adjusted it.
- The essential role of cholesterol in SARS-CoV-2 replication has recently been observed. How does this relate to omicron? What role does cholesterol have in entry and replication?
Response: Thank you very much for your comment. Cholesterol in the viral envelope and cell membranes contributes to the replication of several viruses, including SARS-CoV-2, acting as a key component in viral entry. Cholesterol-rich lipid rafts represent a surface through which viruses enter the cell through endo-cytosis and are also essential for the interaction between protein spike and the angiotensin converting enzyme 2 receptor. The impact of cholesterol on coronavirus infectivity was supported by research into the effect of cholesterol depletion on SARS-CoV infection, which led to a significant reduction in viral mRNA. Cholesterol depletion has been shown to reduce viral entry and virus-induced fusion. The implications of cholesterol in replicating the virus for the new Omicron strain, which is a variant of SARS-CoV-2, are the same.
- Are there risks involved in taking medication (the drugs included in this review) for the treatment of COVID-19? Are these drugs specifically targeting viral replication or are they potentially inhibiting other stages of infection or other host pathways that the virus requires?
Response: Thank you very much for your comment. The administration of these medicines may involve certain risks, especially for people who have comorbidities and who use other medicines. For example, using the new oral antiviral at the same time as other drugs can lead to considerable drug interactions. Thus, it is not recommended that Paxlovid be administered with drugs that induce the same enzymes, because they may metabolize nirmatrelvir or ritonavir too quickly, and the virologic response may be lost and viral resistance may develop. In addition, ritonavir may cause liver damage and therefore should be avoided in patients with liver disease.
Most drugs used in the treatment of SARS-CoV-2 target viral replication. A clear example is remdesivir. Remdesivir is an adenosine analogue that interferes with the function of the RNA-dependent RNA polymerase enzyme and prevents the genetic modification of the virus by the exoribonuclease enzyme, reducing the production and replication of the virus. The same target is followed by chloroquine and hydroxychloroquine, which inhibit the glycosylation of ACE 2, prevent spike protein binding and induce replication inhibition and virus release.
- The authors have described various drugs that are currently approved for the use in humans for other diseases may need to be listed in a table. In addition, these drugs may have side effects, which may need to be described and provide opinion on which of them have lesser side effects. That opinion may matter in selection of drugs to screen for anti-SARS-CoV-2 omicron variant.
Response: Thank you very much for the suggestion. We have added a table with the main molecules discussed in this review in which we have pointed out the pathologies for which they were initially approved together with the mechanism of action
In each therapeutic class, the most common side effects are highlighted, which may differ depending on the particularities of the individual. Adverse reactions range from gastrointestinal disorders (remdesivir, lopinavir / ritonavir, ribavirin, chloroquine / hydroxychloroquine), changes in lipid and carbohydrate metabolism (lopinavir / ritonavir, corticosteroids, tocilizumab), liver damage (remdesivir, tocilizumab) and up to haematological disorders (ribavirin) and allergic reactions (tocilizumab).
We hope you find the revised manuscript acceptable for publication. Thank you once again for your consideration.
Best regards,
Alexandra Scurtu,
Timisoara,
31th of January, 2022

Round 2
Reviewer 1 Report
Good job.Only revise the grammer and punctuation
Author Response
Thank you very much for the time allocated for the review of the manuscript “The “invisible enemy” SARS-CoV-2: viral spread and drug treatment” as well as for your pertinent observations.
Please see below a point-by-point response to your comments:
Good job. Only revise the grammar and punctuation
Response: Grammar and punctuation have been revised.
We hope you find the revised manuscript acceptable for publication. Thank you once again for your consideration.
Best regards,
Alexandra Scurtu,
Timisoara,
5th of February, 2022

Reviewer 2 Report
Most of the suggestions have been addressed and the work has been improved substantially.
In my opinion, the work is suitable for publication after introducing only minor changes, i.e .:
a / figure 3: in paper, there are two versions of the same figure, both are still of poor quality. Before publication, consider deleting this figure and replace it with a new one of better quality.
b / figure 5: in my opinion, it still requires a more detailed description.
Author Response
Thank you very much for the time allocated for the review of the manuscript “The “invisible enemy” SARS-CoV-2: viral spread and drug treatment” as well as for your pertinent observations.
Please see below a point-by-point response to your comments:
a / figure 3: in paper, there are two versions of the same figure, both are still of poor quality. Before publication, consider deleting this figure and replace it with a new one of better quality.
Response: Thank you very much for your comment. We have changed it.
b / figure 5: in my opinion, it still requires a more detailed description.
Response: Thank you very much for the suggestion. We have adjusted it.
We hope you find the revised manuscript acceptable for publication. Thank you once again for your consideration.
Best regards,
Alexandra Scurtu,
Timisoara,
5th of February, 2022
